



# Model-driven optimization of coastal sea observatories through data assimilation in a finite element hydrodynamic model (SHYFEM v. 7_5_65)

Christian Ferrarin[1], Marco Bajo[1], and Georg Umgiesser[1,2]

[1]CNR - National Research Council of Italy, ISMAR - Marine Sciences Institute, Venice, Italy
[2]Marine Research Institute, Klaipeda University, Klaipeda, Lithuania

**Correspondence:** Christian Ferrarin (c.ferrarin@ismar.cnr.it)

**Abstract.** Monitoring networks aims at capturing the spatial and temporal variability of one or several environmental variables in a specific environment. The optimal placement of sensors in an ocean or coastal observatory should maximize the amount of collected information and minimize the development and operational costs for the whole monitoring network. In this study, the problem of the design and optimization of ocean monitoring networks is tackled throughout the implementation of data
assimilation techniques in the Shallow water HYdrodynamic Finite Element Model (SHYFEM). Two data assimilation methods - Nudging and Ensemble Square Root Filter - have been applied and tested in the Lagoon of Venice (Italy), where an extensive water level monitoring network exists. A total of 29 tide gauge stations were available and the assimilation of the observations result in an improvement of the performance of the SHYFEM model that went from an initial root mean square error (RMSE) on the water level of 5.8 cm to a final value of about 2.1 and 3.2 cm for the two data assimilation methods, respectively.
In the monitoring network optimization procedure, by excluding just one tide gauge at a time, and always the station that contributes less to the improvement of the RMSE, a minimum number of tide gauges can be found that still allow for a successful description of the water level variability. Both data assimilation methods allow identifying the number of stations and their distribution that correctly represent the state variable in the investigated system. However, the more advanced Ensemble Square Root Filter has the benefit of keeping a physically and mass conservative solution of the governing equations, which
results in a better reproduction of the hydrodynamics over the whole system. In the case of the Lagoon of Venice, we found that, with the help of a process-based and observation-driven numerical model, two-thirds of the monitoring network can be dismissed. In this way, if some of the stations must be decommissioned due to a lack of funding, an a-priori choice can be made, and the importance of the single monitoring site can be evaluated. The developed procedure may also be applied to the continuous monitoring of other ocean variables, like sea temperature and salinity.

# 1  Introduction

Ocean and coastal monitoring networks are fundamental for tracking contaminants in the water, assessing environmental change and water quality, observing sea level rise and developing strategies for managing resources in a changing climate (Stammer et al., 2019; Trowbridge et al., 2019). Coastal zones are dynamic and subject to changing environmental conditions





caused by natural and anthropogenic variations in climatic and oceanographic processes. The monitoring of the spatial and temporal complexity of the coastal ocean is challenging and a large number of observational sites are required to correctly describe the interactions at the land-sea transition, and coupled physical, chemical, and biological processes. However, the implementation and maintenance of such large monitoring networks are expensive and therefore their optimization is of crucial importance. In the last decades, satellite earth observation technologies have been widely used to integrate in-situ observatories for better understanding the current state of oceans and coastal seas (Levy et al., 2018).

Oceanographic models are increasingly used in coastal systems to describe sea dynamics induced by tide, atmospheric and terrestrial forcing, complementing thus the collected information retrieved by direct observations (Mey-Frémaux et al., 2019). Numerical models are also often used for predicting the ocean conditions, especially during storm events for endangered areas (Chaumillon et al., 2017). All models, however, need observations of the sea state to be calibrated and validated. Once the model is calibrated, new measurements can be used in a continuous validation of the model results. Observations can also be assimilated into the model, increasing its capacity to represent the dynamics of the investigated system (Edwards et al., 2015; Carrassi et al., 2018). In this case, we can speak of observations that improve the numerical model.

There is however another point of view. If only observations would be available, the best distribution of the monitored variable over the system could be given only by data interpolation (DI) of the observation points to the other areas. The direct observations of the sea conditions are considered to represent the true state at the monitoring point. However, the spatio-temporal interpolation of such true values is not meant to correctly describe the variability of the investigated state variable over the whole system. This is especially true in the coastal systems that are characterized by complex small scale and high-frequency dynamics. In this case the resulting picture of interpolated values may show non coherent features and inconsistency between data points. When an oceanographic model is available, the interpolation of these observations can be carried out by the model and much better representation of the environment can be achieved. In this contest, models are used to connect sparsely (in space and time) observations or synthesizing them through data assimilation (DA) techniques (Mey-Frémaux et al., 2019).

Validated ocean circulation model and DA can also assist the network design of a new observing system or optimizing existing observatory (Fujii et al., 2019). In the case of new monitoring networks, Observing System Simulation Experiments (OSSEs) are performed assimilating synthetic observation data (generated from a free-running model simulation that is intended to represent a virtual "true" ocean) into other data-assimilative simulation runs in which different initial/forcing conditions are used (Raicich, 2006; Xue et al., 2011). The evaluation of the impact of the assimilated data in the OSSE simulations allows designing an optimal observing system. In order to evaluate existing monitoring networks, Observing System Experiments (OSEs) are performed by assimilating in several simulations a certain amount or type of observations and evaluating their impacts on the model against a reference dataset. Such an approach can be adopted in coastal regions to optimize existing observational arrays, with implications on sampling technology and networks (Frolov et al., 2008; Schulz-Stellenfleth and Stanev, 2010).

In this study, we show how data assimilation techniques are implemented in the Shallow water HYdrodynamic Finite Element Model (SHYFEM) for optimizing the tide gauge network of the Lagoon of Venice (Italy). Since one limitation of the





observing system evaluation procedure is that it depends on the properties of the DA employed for the evaluation (Fujii et al.,

2019), here we adopted a multiple systems approach implementing the Nudging and the Ensemble Square Root Filter data assimilation methods.

## 2   Methods

### 2.1   SHYFEM model description

The numerical experiments consisted of simulating the circulation in the Lagoon of Venice using the open-source SHYFEM

hydrodynamic model (Umgiesser et al., 2014). The model has been already applied to simulate hydrodynamics in the Mediterranean Sea (Ferrarin et al., 2018), in the Adriatic Sea (Bellafiore et al., 2018; Bajo et al., 2019) and in several coastal systems (Umgiesser et al., 2014, and references therein). The model solves the shallow-water equations in their formulations with levels and transports using a finite-element numerical method and semi-implicit time stepping. In the present work, a relatively simple two-dimensional configuration of the model has been used, solving the following equations:

$$\frac{dU}{dt} - fV = -H\left(g\frac{\partial \zeta}{\partial x} + \frac{1}{\rho_w}\frac{\partial p_a}{\partial x}\right) + A_H\nabla^2 U + \frac{1}{\rho_w}\left(\tau_{wx} - \tau_{bx}\right) \tag{1a}$$

$$\frac{dV}{dt} + fU = -H\left(g\frac{\partial \zeta}{\partial y} + \frac{1}{\rho_w}\frac{\partial p_a}{\partial y}\right) + A_H\nabla^2 V + \frac{1}{\rho_w}\left(\tau_{wy} - \tau_{by}\right) \tag{1b}$$

$$\frac{\partial \zeta}{\partial t} + \frac{\partial U}{\partial x} + \frac{\partial V}{\partial y} = 0 \tag{1c}$$

where $t$ is the time, $x$ and $y$ are the spatial Cartesian coordinates and $\eta = \eta(x,y,t)$ is the water level. $U = U(x,y,t)$ and $V = V(x,y,t)$ are the zonal and meridional water transport components, $g$ is the acceleration due to gravity, $pa = pa(x,y,t)$ is the atmospheric pressure at mean sea level, $\rho_q$ the average density of sea water, $h = h(x,y)$ is the water depth at rest, while $H = h + \eta$ is the total water depth and $f = f(y)$ is the Coriolis parameter, varying with latitude. Smagorinsky's formulation (Smagorinsky, 1963; Blumberg and Mellor, 1987) is used to parameterize the horizontal eddy viscosity ($A_h$). $\tau_{wx}$ and $\tau_{wy}$ are

the two components of the wind stress in the $x$ and $y$ directions and $\tau_{bx}$ and $\tau_{by}$ are the two components of the bottom stress.

The Coriolis term and pressure gradient in the momentum equation, and the divergence terms in the continuity equation are treated semi-implicitly. Bottom friction and vertical eddy viscosity are treated fully implicitly for stability reasons due to the shallow nature of the lagoon, while the remaining terms (advective and horizontal diffusion terms in the momentum equation) are treated explicitly. A detailed description of the model equations is given in Umgiesser et al. (2014) and Bellafiore et al.

85   (2018).





## 2.2 Data assimilation methods

### 2.2.1 Nudging

The nudging method is a flexible assimilation technique that is computationally more economical than other assimilation methods like variational data assimilation. First used in meteorology (Hoke and Anthes, 1976), the nudging method has been used with success in modelling the atmosphere (Stauffer and Seaman, 1990) and in oceanography (Verron, 1990; Blayo et al., 1994). Nudging is a simple assimilation technique where a new source term is added to the prognostic equations that drag the results versus the observed values. Therefore, it uses dynamical relaxation of the equations to tend to the observational points. The extra term to be introduced in the prognostic equation can be formulated as:

$$\partial S/\partial t = ... + (S_{obs} - S)/\tau \qquad (2)$$

where $S$ is the variable where nudging has to be applied, $S_{obs}$ the observation value, and $\tau$ is the relaxation time scale. Depending on the value of $\tau$, the relaxation is very strong (small $\tau$) or weak (large $\tau$). The value of $\tau$ can be different from point to point. It is worth mentioning that, by adding this extra term in the governing equations (e.g. the continuity Eq.1c for the water level), the numerical solution is no more mass conservative.

### 2.2.2 Ensemble Square Root Filter

The ensemble square root filter (hereinafter referred to as EnSRF) is a more complex assimilation method, widely used in environmental sciences (Evensen, 2004), and can be considered as an evolution of the Ensemble Kalman Filter (EnKF, Evensen, 2003). The assimilation code that allows one to use both these methods, has been recently implemented in SHYFEM (https://github.com/marcobj/shyfem) and the code was used for the first time in a study on seiches and storm surges in the Adriatic Sea (Bajo et al., 2019).

The formulation of the EnSRF is slightly different from the EnKF and avoids the perturbation of the observations. Using the notation of Evensen (2004), if we define the model states as $\psi_i \in \mathbb{R}^n$ and the matrix holding them as:

$$\mathbf{A} = (\psi_1, \psi_2, ..., \psi_N) \in \mathbb{R}^{n \times N}, \qquad (3)$$

with $N$ the number of ensemble members and $n$ the dimension of the states, the mean is:

$$\overline{\mathbf{A}} = \mathbf{A} \mathbf{1}_N, \qquad (4)$$

where $\mathbf{1}_N \in \mathbb{R}^{N \times N}$ is a matrix with each element equal to $1/N$. The ensemble approximation of the background error covariance, $\mathbf{P}$, is:

$$\mathbf{P}_e = \frac{\mathbf{A}'(\mathbf{A}')^T}{N-1}, \qquad (5)$$





where $\mathbf{A}' = \mathbf{A} - \overline{\mathbf{A}}$, is the matrix containing the ensemble perturbations. The covariance update in the Kalman filter is:

$$\mathbf{P}^a = \mathbf{P}^f - \mathbf{P}^f \mathbf{H}^T (\mathbf{H} \mathbf{P}^f \mathbf{H}^T + \mathbf{R})^{-1} \mathbf{H} \mathbf{P}^f, \tag{6}$$

with the index $a$ as analysis, $f$ as first guess, $\mathbf{H} \in \mathrm{I\!R}^{m \times n}$ the observation operator, with $m$ the number of observations, and $\mathbf{R}$ the observation error covariance matrix. In the ensemble methods, this equation is written as:

$$\mathbf{A}^{a\prime} \mathbf{A}^{a\prime T} = \mathbf{A}'(\mathbf{I} - \mathbf{S}^T \mathbf{C}^{-1} \mathbf{S}) \mathbf{A}'^T$$
$$\mathbf{S} = \mathbf{H} \mathbf{A}' \tag{7}$$
$$\mathbf{C} = \mathbf{S}\mathbf{S}^T + (N-1)\mathbf{R}.$$

After some eigenvalue and singular value decompositions (see the paper Evensen, 2004), the equation splits into two symmetrical parts:

$$\mathbf{A}^{a\prime} \mathbf{A}^{a\prime T} = \left(\mathbf{A}' \mathbf{V}_2 \sqrt{\mathbf{I} - \boldsymbol{\Sigma}_2^T \boldsymbol{\Sigma}_2}\right) \left(\mathbf{A}' \mathbf{V}_2 \sqrt{\mathbf{I} - \boldsymbol{\Sigma}_2^T \boldsymbol{\Sigma}_2}\right)^T, \tag{8}$$

where $\mathbf{V}_2 \in \mathrm{I\!R}^{N \times N}$, $\boldsymbol{\Sigma}_2 \in \mathrm{I\!R}^{m \times N}$ are two matrices coming from the decomposition of $\mathbf{S}^T \mathbf{C}^{-1} \mathbf{S}$ and $\mathbf{I}$ is the identity matrix. The solutions are:

$$\mathbf{A}^{a\prime} = \mathbf{A}' \mathbf{V}_2 \sqrt{\mathbf{I} - \boldsymbol{\Sigma}_2^T \boldsymbol{\Sigma}_2} \boldsymbol{\Theta}^T, \tag{9}$$

for any random orthogonal matrix $\boldsymbol{\Theta}^T$. This allows a random redistribution of the variance reduction among the ensemble members.

The approximation of the covariance matrix with the ensemble members (eq. 5) becomes perfect when $N$ goes to infinity. However, with a finite number of ensemble members, model variables that are far and not really correlated, can have have a variance different from zero. To avoid this issue, keeping a reasonable number of ensemble members, we apply a localisation scheme. Localisation is often used in ensemble data assimilation and can be done following different methods (Houtekamer and Mitchell, 2001; Hamill et al., 2001; Anderson, 2003). In the present case we used a local analysis method, which performs in a similar way of the covariance localisation method (Sakov and Bertino, 2011).

This method reduces the influence of the observations too far from the location of a model variable. If the model has $N$ variables, the distance of each of them from each observation is computed and a weighting factor, depending on such distance, is computed. We used a Gaspari-Cohn function (Gaspari and Cohn, 1999), with which the weight decreases in a way similar to a Gaussian, but vanishes for distances $r > 2d$, where $d$ is a *cut-off* distance. Instead of making a global analysis, the analysis is made for each node of the grid and the matrices are reduced to a local dimension. Then, the total analysis is the sum of all the local contributions (Carrassi et al., 2018).





## 2.3 The optimization procedure

Starting from the DA run with the assimilation of all stations ($N$), the monitoring network evaluation procedure was designed

as an iterative process in which several numerical simulations are carried out excluding one tide gauge from the assimilation at a time. In this study, we consider the root mean square error (RMSE) of the simulated values respect to the observations as the cost function to be minimized in the optimization process. Similar to the approach described in the previous section, for each run the RMSE is evaluated for all data points. After doing this for all remaining stations, the observation site that contributes less to the improvement of the RMSE (the one having the lowest RMSE value) is excluded in the next optimization step

(assimilation of $N-1$ stations). The iterative process continues ($N-2, N-3, N-4, ...$) until only one station is assimilated. At each optimization step, the mean RMSE over the whole monitoring network is evaluated. The whole optimization procedure requires $N \times (N+1)/2$ numerical simulations. In the case of the DA-EnSRF, the computational effort is much higher and depends on the number of members of the ensemble.

The optimization procedure is easily and efficiently parallelized since all simulations within each iteration step are indepen-

dent of each other. Similarly, all members of each DA-EnSRF process are independent and can be carried out simultaneously on different processors.

## 2.4 Application to the Lagoon of Venice

The Lagoon of Venice (Fig. 1) is situated in the Northern Adriatic Sea and is the largest Mediterranean lagoon (area of 550 km$^2$). The principal hydraulic forcings of the Lagoon of Venice are the tide and the wind Umgiesser et al. (2004b). Even if

the lagoon is a micro-tidal system (tidal range of about 80 cm), tides are a major factor in shaping landforms and driving ecological gradients and biological communities. The lagoon is separated from the open sea by barrier islands, and three inlets (Lido, Malamocco, and Chioggia) ensure an active renewal of the lagoon waters (Ferrarin et al., 2017). The lagoon is characterized by a complex system of tidal channels. The density of the drainage network increases landward as main tidal collectors departing from the inlets branch in progressively smaller-size channels, ranging in depth from a more than 15 m of

main reaches to few decimetres of salt marsh creeks (Madricardo et al., 2017). Such a drainage network cuts across a large extent of shallow water areas, which have an average depth of 1 m and include mudflats and salt marshes.

The city of Venice is located in the centre of the lagoon and is composed of more than a hundred islands linked by bridges. The elevation of these islands is extremely low, subjecting them to flooding during storms, which in turn threatens the unique cultural heritage of this city and affects its everyday life. The northern Adriatic Sea is frequently affected by storm surge events,

mainly triggered by strong south-easterly wind (Orlić et al., 1994). It is therefore of crucial importance for the management of this environment to monitor water level variations outside and inside the lagoon.

### 2.4.1 The tide gauge network

The Lagoon of Venice has two tide gauge networks for supporting the local real-time storm surge prediction and warning system. They are managed by the Institute for Environmental Protection and Research - National Centre for Coastal



Zone and Characterization Marine Climatology and for Operational Oceanography (ISPRA, Unit for Tides and Lagoons, http://www.venezia.isprambiente.it/, last access 10 January 2020) and the Tide Forecast and Early Warning Center of the City of Venice (CPSM, https://www.comune.venezia.it/it/content/centro-previsioni-e-segnalazioni-maree, last access 10 January 2020). ISPRA manages a network of 45 tide gauge stations equipped for the systematic measurement of water level and other related parameters, such as wind direction, wind speed, atmospheric pressure, precipitation, and wave-height inside the Lagoon

of Venice and in the north-western Adriatic coastline. The monitoring network of CPSM consists of 17 hydro-meteorological stations distributed within the lagoon and along the Venetian littoral for the real-time monitoring of the water levels, waves and meteorological parameters. Some locations with high valuable relevance are monitored by both institutions.

In this study, we collected all the available data from both the ISPRA and CPSM monitoring networks over a one-month period (November 2013) with the highest number of stations without missing data. The selected dataset consists of quality-

controlled 10-minute values of sea level measured at the 29 tide gauge stations marked with red dots in Fig. 1. As shown in the figure, all tide gauges are installed within navigational channels in order to allow their installation and maintenance. Most of the tide gauges are located in the central and northern parts of the lagoons, where most of the urban settlements are placed (Venice, Murano and Burano), at the inlets and in the southern end of the lagoon near Chioggia. The selected period of investigation comprises both calm weather conditions as well as significant wind events.

In order to investigate at which degree the observations represent the state variable over the whole system, a field approximation through optimal interpolation (OI) of the data have been performed. OI is a commonly used and fairly simple method to perform interpolation of sparse data and also in data assimilation. OI was first described in Gandin (1965) and other references and implementations can be found also in Daley (1991). It is also often referred to as statistical interpolation. In OI, starting from a background grid, observation points are used to correct the background grid. Points that lie close to each other are given

less weight. The interpolation of the water levels was carried out on a $0.5 \times 0.5$ km regular grid.

### 2.4.2 Simulation set-up

The water circulation in the Lagoon of Venice, induced by tide and wind was simulated by the unstructured model SHYFEM applied over a spatial domain that represents the entire Lagoon and its adjacent shore. The model adequately reproduces the complex geometry and bathymetry of the Lagoon of Venice using unstructured numerical meshes composed of triangular

elements of variable form and size, going down to a few meters in the channels (Fig. 1). The model bathymetry was obtained from the data collected in 2002 by Magistrato alle Acque di Venezia - merged with later surveys - and the 2014 MBES bathymetry acquired by CNR-ISMAR in the main channels of the lagoon (Madricardo et al., 2017).

The application of the SHYFEM model to the Lagoon of Venice has been validated in previous works reproducing correctly tidal propagation, storm surge, water flows at the lagoons' inlets and water temperature and salinity variability (Umgiesser

et al., 2004a; Ferrarin et al., 2008, 2010; Ghezzo et al., 2011).

In this study, hydrodynamics in the lagoon was simulated using 10-minute observed forcing and boundary conditions (i.e., wind stress and open sea level). The initial condition is always a calm state. This is certainly no problem for the current velocity





and the water level since these quantities approach a dynamic state very fast (less than a day). The numerical simulations were performed over the period covered by the selected dataset (November 2013).

In order to apply the Nudging DA method, a value for the relaxation parameter $\tau$ has to be determined. In our case, it was supposed that every observation point would only influence the grid points up to a certain distance. For every observation, a Gaussian bell curve was constructed. The standard deviation of the curve ($sigma$) was set to 2 km, and all points further than 3 standard deviations are excluded from the computations (Fig. 2a). Overlapping areas of influence are considered by summing the value of the Gaussian curve in these points. The $\tau$ value at the peak point of the Gaussian curve was set to 100 seconds, and this value then increases smoothly to infinity in order to simulate an influence which becomes lower when moving away from the observation point.

The EnSRF needs an ensemble of model states that should ideally represent the error of the simulation. In the present case the ensemble of the model states is created varying the boundary condition. We used 60 perturbations for the sea-level boundary condition, with a zero mean and a standard deviation set to 30 cm. This value was found empirically, in order to have a good spread at the boundary, which is then propagated to the variables computed by the model. As asserted, the perturbations are centred and correlated in time with a decay time of two days (red noise). We made also perturbations for the wind, with the same method, but using a standard deviation proportional to 40% of the wind speed. Due to the small study area, we considered the wind constant in space so that the perturbations can vary only in time, as the boundary conditions. However, because of the smallness of our system, the perturbations on the wind are not very effective, as well as perturbations on the initial state. Therefore, the perturbations at the boundary condition are necessary both to create the initial ensemble of states and to keep the spread of the ensemble during the whole time of the simulation.

We run also several preliminary tests to empirically found the best cut-off distance for the local analysis, which was fixed to 0.1 geographical degrees (about 10 km). In order to illustrate the important effect of the localisation, in Fig. 2b we show the correlation values between each observation station and each model level in each node of the model grid, at a specific time-step. The correlation is weighted with the Gaspari-Cohn function, which vanishes the values too far from the station. This quantity is not used directly by the local analysis routine, but it is useful to understand its effect. Note also that this is the correlation with the water levels, but the EnSRF considers also the cross-correlations with the water velocities and corrects them as well. The strong difference with the relaxation time used by the nudging to weight the observations (Fig. 2a), is that the use of the real correlations between the model variables produces an anisotropic distribution of the observation correction, which respects the water dynamics forced by the channels, by the tidal flats and by the basin morphology. Moreover, as the dynamics varies at each time-step, so does the correlation between model variables and the weight of the assimilation increments.

The EnSRF assimilates water level from the selected stations considering them independent (the $R$ error covariance matrix is diagonal) and the error of each station is set to 1 cm. The model evolves forward in time the ensemble members, each one with different boundary condition and wind forcing, and an analysis step is done every hour. The results considered in this work are extracted by the analysis states, which are saved every hour.





## 3   Results

In the exposition of the results, we defined the model run without data assimilation as the control simulation, while, for both the DA schemes, the base run accounts for the assimilation of all the 29 monitoring stations. All mentioned parameters ($tau$, $sigma$, cut-off distance for the local analysis) were manually defined through trial and error calibration process and evaluating the goodness-of-fit of the water level RMSE in the DA-Nudging and DA-EnSRF base simulations.

### 3.1   Data interpolation vs. data assimilation

When entering a shallow basin, as the Venice lagoon, the tidal wave is deformed, either damped or amplified, according to a relationship between local flow resistance and inertia, and the characteristics of the incoming tidal wave (Ferrarin et al., 2010). In the data interpolation method, the distribution of the water levels is given by a spatial interpolation of the observations. Fig. 3a reports a snapshot of the interpolated water levels over the lagoon during a flood tide. The map shows, for this particular time frame, a patchy non coherent distribution with the lowest values in the nearshore area close to the inlets, while the highest are in proximity of the central and northern lagoon's margins.

Does the interpolation of the observations provide a realistic spatial representation of the water level variability over the lagoon domain? To answer to this question, we show in Fig. 3b the water level computed by the model, without any data assimilation (Control sim.). The nudging run (DA-Nudging) is shown in Fig. 3c and the EnSRF run (DA-EnSRF) in Fig. 3d. The control simulation has a completely different distribution of the water levels with respect to the data interpolation. The mode simulation shows the lowest water level in the open sea, which gradually increases going from the inlets to the inner lagoon, describing the propagation of the tidal wave. The three inlets lead the water circulation in three sub-basins, divided by narrow areas with little water exchange (these zones are identified as dynamical watersheds). The modelled maps (control, DA-Nudging and DA-EnSRF) clearly account for islands and marsh boundaries. DA-Nudging shows a similar representation of the control simulation, but with slightly higher values of the water levels on the central and southern tidal flats (Fig. 3c). Similarly, the DA-EnSRF adjusts the water levels towards the observations keeping the physical dynamics of the flow (Fig. 3d). It is worth mentioning that the water level distributions at different tidal phases would lead to similar DI and DA considerations.

In order to establish which method better represents the water level variability over the lagoon, we need to evaluate the capacity of each approach to describe the parameter at locations not included in the computation. Thanks to a large number of available tide gauges in the Lagoon of Venice, the model skill assessment (in terms of the root mean square error, RMSE) is determined by re-running DI and DA experiments removing one station from the assimilation and comparing the water level in this station with the modelled one. The evaluation procedure was repeated for each monitoring station and the results are reported in Table 1. When using the optimal interpolation approach, the average RMSE is 3.9 cm, with values ranging from 0.8 to 8.5 cm. The highest RMSE is found at stations located at the lagoon margins (9, 14, 25 and 27) and the Chioggia and Malamocco inlets (4 and 12). The control SHYFEM simulation, the one without data assimilation, has a mean RMSE of 5.8 cm, with the highest errors found at the stations located near the lagoon margins (1, 9, 14, 24 and 29). The correlation coefficient (not reported in the table) is everywhere higher than 0.97, except for station 24 where it is 0.47. Therefore, from the



statistics we deduce that the control simulation has a worse performance with respect to the direct interpolation of the data and that it slightly fails in reproducing correctly the water dynamics in border areas, especially in the small creeks surrounded by mashes (e.g. station 24). However, even if data interpolation is statistically better, looking at Fig. 3a the spatial distribution of the water level is clearly unphysical.

Differently, both DA methods strongly improved the model skills in all parts of the lagoon. The average RMSE resulted in 2.1 cm and 3.2 cm for DA-Nudging and DA-EnSRF, respectively. The results reported in Table 1 show that results improved at all stations, even those affected by the highest errors in the control simulation. The capacity of the different methods in reproducing the temporal evolution of the water level is shown in Fig. 4 for the station n. 12 (in this case removed from the assimilation/interpolation). It is evident that in this case, the interpolation does not represent correctly the water level variability, being influenced by values recorded outside the lagoon domain, which do not take into account the correct tidal propagation dynamics. On the other hand, the data assimilation results adjust the water levels towards the observations keeping the physical dynamics of the flow. Therefore, the model simulation with a DA scheme is the approach that better represents the variability of the water levels in the lagoon.

Additionally, in a multivariate analysis approach we tested the capability of the applied DA-driven simulations in reproducing the current velocities recorded by an acoustic Doppler current profiler (ADCP) mounted on the bottom of the Lido inlet, close to the station n. 15 shown in Fig. 1. Time series of observed and simulated vertically integrated velocities are illustrated in Fig. 5, while the statistical results are summarized in Table 2. Since the DA-Nudging does not adjust the velocities according to the correction of the water level, the model computes spurious velocities according to the pressure gradients generated by the water level increments. The simulated current velocities - and therefore the water exchange through the inlets - using DA-Nudging resulted to be higher and slightly out of phase than the observations. Interesting, the DA-Nudging performances on the current velocity are even worse than those of the control simulation. On the other hand, since DA-EnSRF uses cross-correlation to propagate the observation correction to the other model variables, the velocities are corrected according to the modification of the levels, towards a better agreement with the ADCP currents. This is a demonstration of the potentiality of a complex DA method, where a correct specification of the cross-correlations in the background covariance matrix allows a correction of model variables even if they are not directly correlated with the assimilated quantities.

### 3.2 Monitoring network optimization

The next step is to use DA methods to find the minimum number of stations - and their distribution - that correctly represent the state variable in the investigated system. The optimization procedure of this tide gauge network, composed by $N = 29$ stations, requires 435 ($N \times (N+1)/2$) numerical simulations. However, the computational cost of the DA-EnSRF is much higher, since the ensemble is composed by 61 members. So in this case the simulations are 26,535, but the scalability is high since the 61 simulations are independent. The results of the water level observatory evaluation are reported in Fig. 6 in terms of the model RMSE as a function of the number of stations considered in the assimilation. For comparison, the same procedure was applied to the data interpolation.





The evaluation procedure allows finding the minimum number of tide gauges for a successful description of the water level in the lagoon. However, the optimization criterion (the RMSE threshold) is arbitrary and may differ for different environments, state variables and monitoring networks. In the present case, we can see that using both DA-Nudging and DA-EnSRF, the

RMSE does not change too much passing from 29 to 10-12 assimilated stations. Even if the EnSRF has an average RMSE higher than the DA-Nudging, the RMSE of the EnSRF has a slower increase with the reduction of the stations. The initial decrease of the RMSE is probably due to the fact that observations have errors, and tide gauges close to each other can provide slightly different data. The EnSRF considers the observation error in the observation covariance matrix, but it is difficult to find the right value and normally the nominal instrument error is used.

Considering the spatial interpolation method, the use of 10 stations has a RMSE comparable to the error of the control simulation. But we have to stress that in this case the spatial representation of the water level is clearly wrong. We should also mention that the model with the assimilation of only three stations give a lower RMSE than DI with all 29 stations, apart from the fact that results are physically more coherent and consistent.

The resulting optimal distributions of the 10 tide gauge stations determined by DA-Nudging and DA-EnSRF are shown in

Fig. 7. In both cases, the optimization procedure selected tide gauges located near the inlets (one each, avoiding redundancy of nearby stations), in some of the islands in the northern part of the lagoon, and stations along the lagoon margins. We therefore can consider that, with the help of DA methods, only 10 of the considered 29 tide gauges are necessary for properly describing the spatial and temporal variability of the water level in the Lagoon of Venice. Considering that the average annual maintenance cost of a tide gauge in the Lagoon of Venice is approximately 3,500 € (Alvise Papa, CPSM, personal communication), the

optimization of the monitoring network could allow saving about 66,000 € per year.

However, the choice of which stations to keep in the monitoring network depends also on many practical factors. As an example, the monitoring authority would decide to keep some stations because of their strategic relevance, maintenance costs, distance from the laboratory or for continuing long-term time series. The optimization method can be easily customized based on predetermined specific constraints. As a realistic exercise, we fixed the stations at the inlets (4, 12, 11) and in the main urban

settlements (2, 6, 17, 19) in the monitoring network. The evaluation procedure is then repeated using the DA-Nudging method, keeping these 7 stations and the results are presented in Fig. 8. In this customized optimization exercise, the results show that 15 stations are necessary to guarantee a proper description of the water level variability in the lagoon.

## 4 Discussion and Conclusions

The methodology presented in this study allows for the evaluation of existing coastal observatories. Using a DA system, which

is an observation-driven and process-based method, the iterative optimization procedure establishes the relevance of each single monitoring station on the description of the considered environment. The example reported in this study describes the optimization of an existing observatory with defined monitoring points. However, the methodology could be applied also to design new monitoring networks. As described by Raicich (2006) and Xue et al. (2011), in an observing system simulation





experiment, synthetic observations are generated by a model run in some locations and then they are assimilated as real

observations. The procedure is similar to a *twin* experiment, a method used to assess the quality of a data assimilation system.

As indicated by Fujii et al. (2019), the goodness of the results of such methods strongly depends on the numerical model applied, on the DA scheme implemented and on the optimization procedure. This effect is evident in the results presented above, where the numerical model performances differ when using a different methodology for assimilating the observations. Moreover, the optimization procedure selected some stations at the lagoon edges, where the RMSE of the control simulation

was the highest. The DA scheme should be selected not only considering the computational cost, but also considering the capacity in reproducing a correct multivariate dynamics of the system. This can be done using observations not assimilated of the same type of the assimilated ones and also observations of other variables of the model (as the ADCP data in our case). In semi-enclosed basins such as lagoon environments, the fluxes through the inlets control the water and the sediment and the nutrient exchanges between the sea and the lagoon, influencing the whole dynamic of the system (Ferrarin et al., 2010). Indeed,

the more advanced EnSRF method improved not only the assimilated water level but also the current velocity, and therefore the fluxes, at the inlet. Therefore, as also outlined by many authors (e.g. Jones et al., 2012; Edwards et al., 2015; Bajo et al., 2019), the description of the coastal sea environment can be improved with the use of a modelling, process-based approach and the use of observations in a complex data assimilation system.

Additionally, as specified at section 2.4.2, the perturbation method implemented in the ensemble data assimilation system

allows the creation of ensemble members that are dynamically consistent and generates realistic correlations in the background-error covariance matrix. These correlation, as well as the covariance matrix, are not constant in time, but vary accordingly with the dynamics induced by the periodic tide and by the non-periodic stress of the wind. In designing or optimizing a monitoring network, such correlation matrix represents a precious source of information which can be used to investigate the area of representativeness of each selected station. To better understand the potentiality of the ensemble data assimilation methods, we

show in Fig. 9 the correlation between the sea level at each station locations with the other nodes of the grid, weighted by the Gaspari-Cohn function. The figure is similar to Fig. 2b but in this case the correlations are averaged over the duration of the whole simulation and considers only the stations selected by the optimization procedure. Even at a first glance, this maps give information about the influence area of each station. These areas do not spread isotropically from the station locations, but they are constrained by the morphology and by the water dynamics, which is considered in the model. This is true not only for the

water level but, as asserted before, also the other variables should benefit by the assimilation of water level observations. Maps similar to that in Fig. 9 can be obtained considering the cross-correlation of the sea level with the water current or with other variables like temperature or salinity, in case of a baroclinic model.

The combination of observations and numerical models is particularly important in coastal regions with scarce monitoring resources. However, to reduce the model error, the applied numerical models must correctly reproduce the complex morphology

of the coastline and the exchange processes between the shelf and the open seas. The processes in such complex systems at the land-sea transition are extremely dynamic and require a holistic approach in which all the hydrological entities (river mouth, salt marshes, lagoons, swamps, coastal sea) should be considered as integral parts of the entire domain of computation. Moreover, due to the complex geometry and morphology of the coastal regions, the numerical models need to be able to represent



hydrodynamic conditions with very high resolution, on the horizontal, vertical and temporal dimensions. With respect to the
above-cited requirement, unstructured models - as the one applied in this study - realise a seamless transition between different
spatial scales for reproducing the coastal-sea interactions, adopting a variable resolution of the mesh elements (Ferrarin et al.,
2018; Kärnä et al., 2018; Maicu et al., 2018; Stanev et al., 2018; Androsov et al., 2019). The applied numerical models need to
be continuously evaluated and upgraded to maintain the highest accuracy.

The model-driven optimization procedure was here applied using hindcast simulations, but it can be also used in an fore-
casting modelling for evaluating the effect of the assimilated data on the predictions (Cummings and Smedstad, 2014; Bajo
et al., 2017). An observation assessment is particularly important when the assimilated data come from different data sources
(e.g., fixed monitoring stations, satellite, radar, gliders), or for a priori estimation of new data sources in an already existing DA
system (Bonaduce et al., 2018). It is crucial in operational oceanography to have a DA scheme keeping the correct physical
description of the dynamics in the investigated environment, without introducing errors that can propagate in time. As indicated
by Fujii et al. (2019), in an operational framework a DA system can also be used as an automatic control system for the quality
of observations.

In the case of the Lagoon of Venice tide gauge network, we demonstrated how numerical models with data assimilation can
play a valuable role in optimizing and designing coastal observatories. The iterative optimization process was based on the
evaluation of the RMSE at the stations not assimilated. It is worth noting that the existing monitoring network can be reduced
by a factor of 2/3 using the tide gauge system in conjunction with a high-resolution numerical model, by means of DA. The
applied methodology is easily exportable to other coastal environments and can be extended to other physical variables.

*Code and data availability.* The SHYFEM hydrodynamic model is open source (GNU General Public License as published by the Free Soft-
ware Foundation) and freely available through GitHub at https://github.com/SHYFEM-model (last access: 10 January 2020). The SHYFEM
code version v. 7_5_65 can be accessed from Zenodo (Umgiesser, 2019). The SHYFEM model v. 7_5_65 with with the data assimilation
code (version ens2.1) is available on Zenodo (Bajo, 2020). The data assimilation code is based on the Geir Evensen's routines, available at
the web-page https://github.com/geirev/EnKF_analysis (last access: 20 January 2020). Configuration files, data and scripts used to run the
models and analyse the results presented in this work are available on Zenodo (Ferrarin et al., 2020).

*Author contributions.* GU conceived the idea of the study with the support of CF. GU developed the optimization procedure and the nudging
data assimilation routines, and MB developed the Ensemble Square Root Filter data assimilation software. CF and MB performed the
numerical simulations. All authors discussed, reviewed and edited the different versions of the manuscript.

*Competing interests.* The authors declare that they have no conflict of interest.



*Acknowledgements.* This work was supported by the Venezia2021 research program funded by the Provveditorato for the Public Works of Veneto, Trentino Alto Adige and Friuli Venezia Giulia, provided through the concessionary of State Consorzio Venezia Nuova and coordinated by CORILA. The authors wish to thank the Tide Forecast and Early Warning Center of the City of Venice and the Italian Institute for Environmental Protection and Research (ISPRA) for providing tide gauge and current velocity data.






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

**Figure 1.** Bathymetry and unstructured mesh of the Lagoon of Venice. The red dots mark the tide gauge monitoring station.



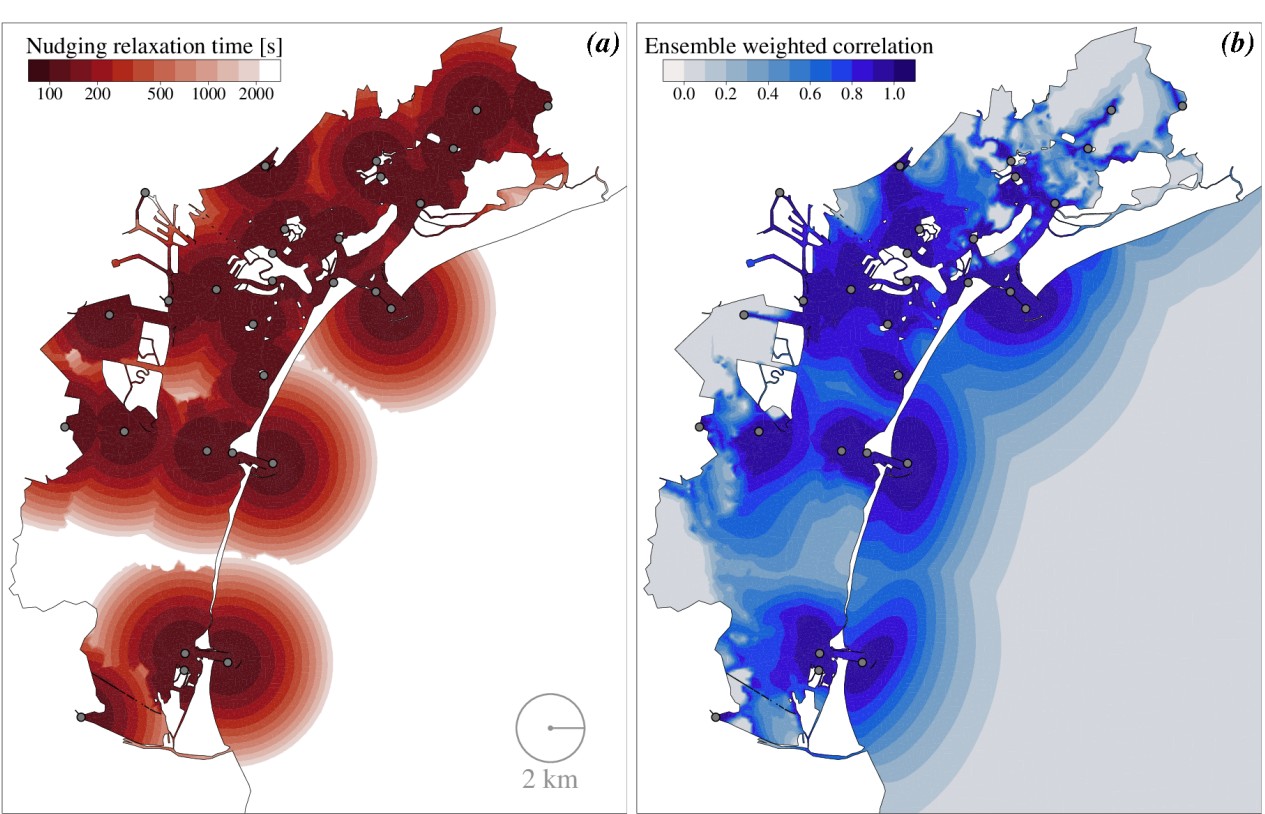

**Figure 2.** Spatial distribution of (a) the relaxation time adopted in the Nudging and (b) the weighted correlation of the ensemble considered in the EnSRF method.



**Figure 3.** A snapshot on 2013-11-04 at 14:00 UTC of the water level distribution in the Lagoon of Venice as obtained by the optimal interpolation (a), the control simulation without assimilation (b), the DA-Nudging base run (c) and the DA-EnSRF base run (d). The gray colour indicates dry salt marshes.

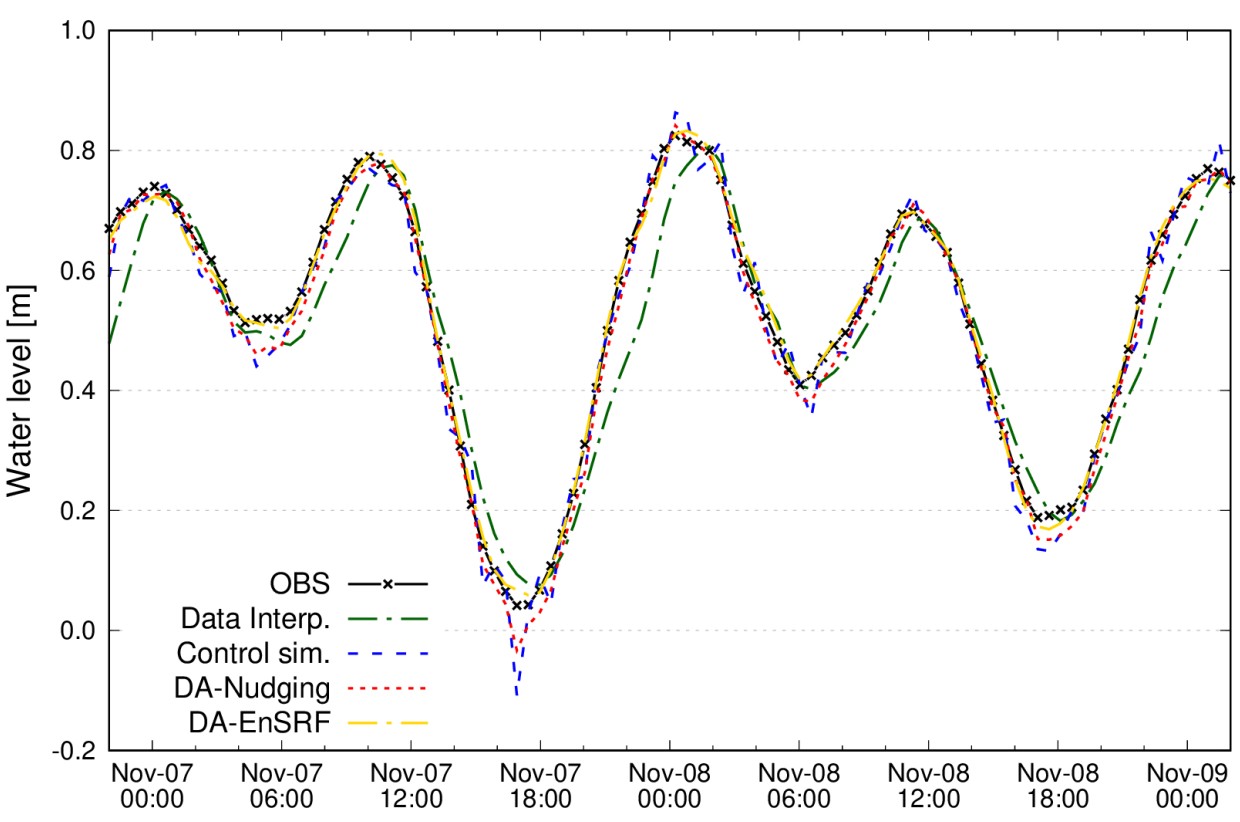

**Figure 4.** Observed, interpolated and simulated water levels at station 12. In this computation, station 12 was not included in the Di and DA algorithms.





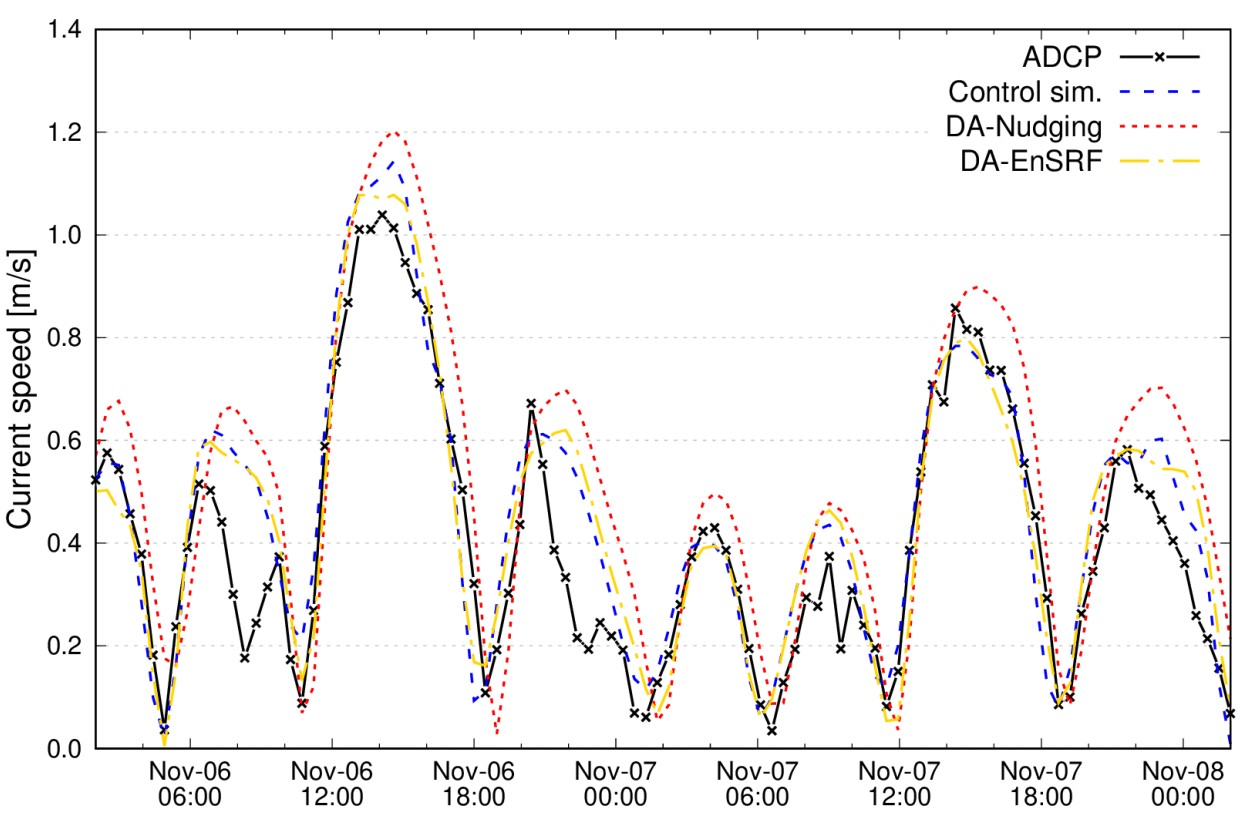

**Figure 5.** Observed and simulated vertically integrated current velocity at the Lido inlet. The ADCP was located close to tide gauge number 15.



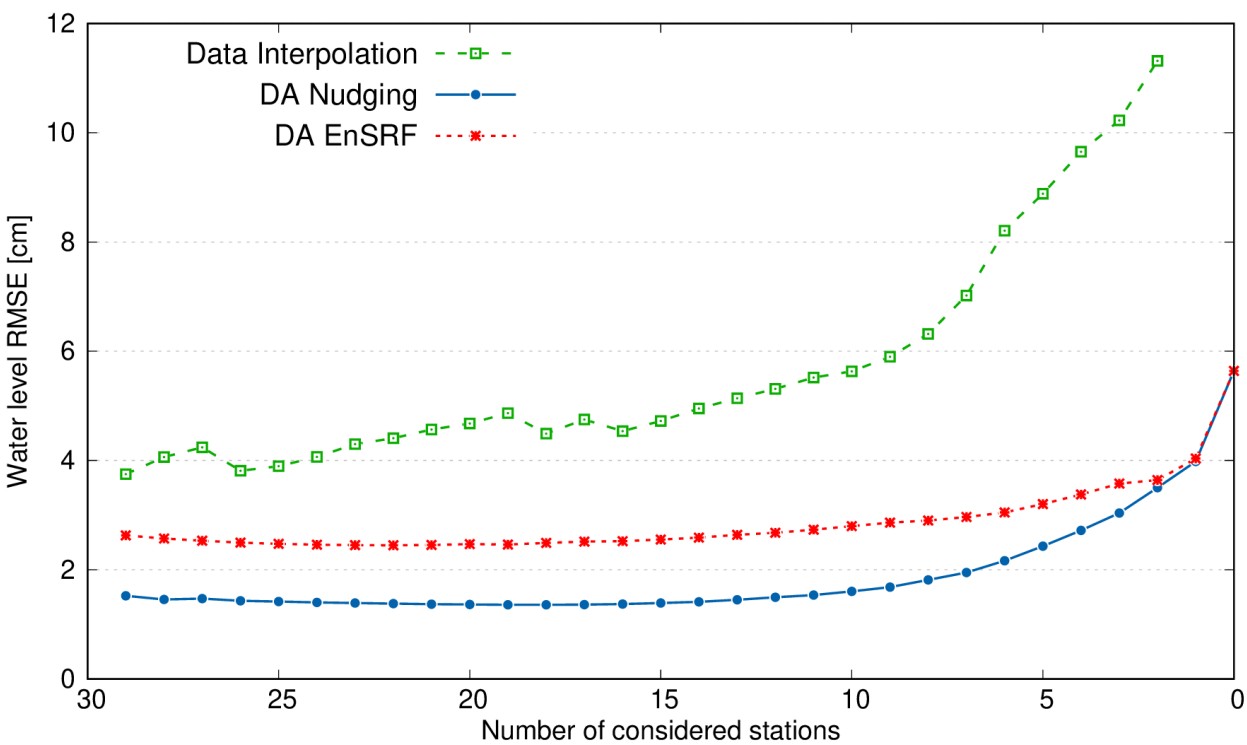

**Figure 6.** Root mean square error of the water levels as a function of the number of tide gauge stations interpolated or assimilated. The RMSE value with zero considered stations for DA is also indicating the error of the base simulation when no DA methods are applied.



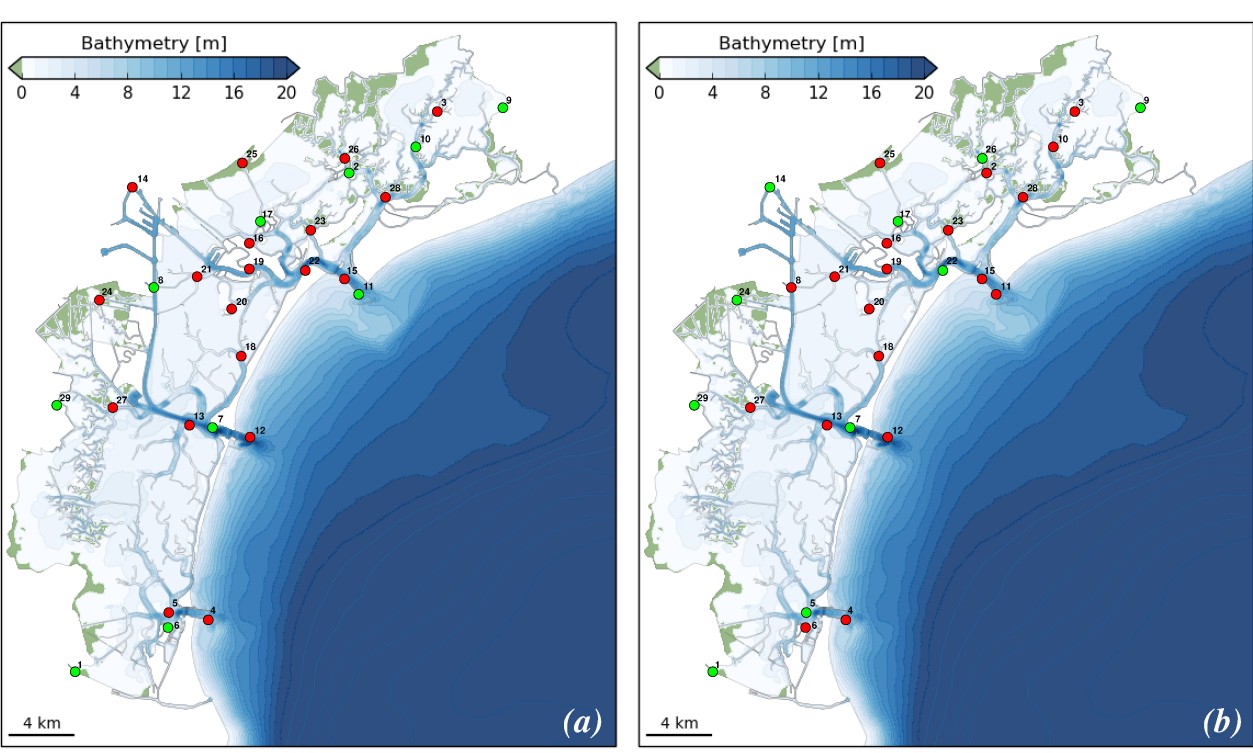

**Figure 7.** The optimal distribution of 10 tide gauge stations (marked with green dots) according to DA-Nudging (a) and DA-EnSRF (b).



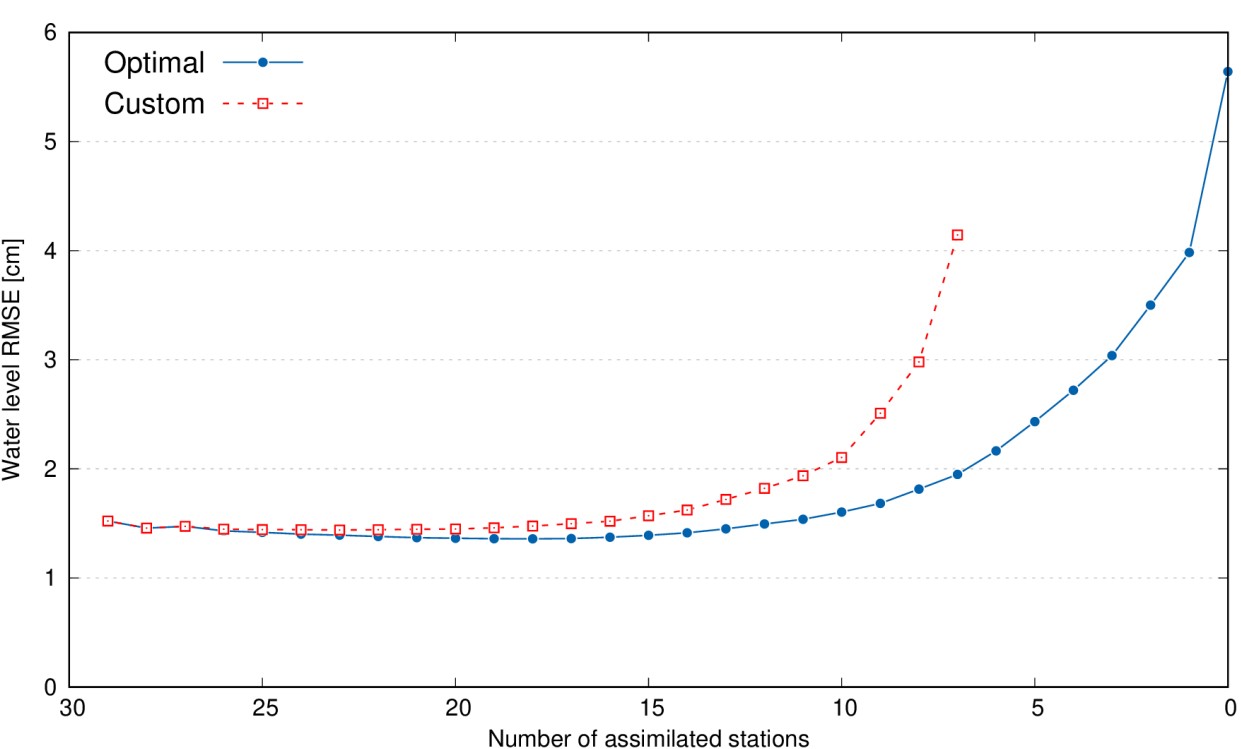

**Figure 8.** Same as Fig. 6, but for the optimal and the custom network experiment using Nudging data assimilation.



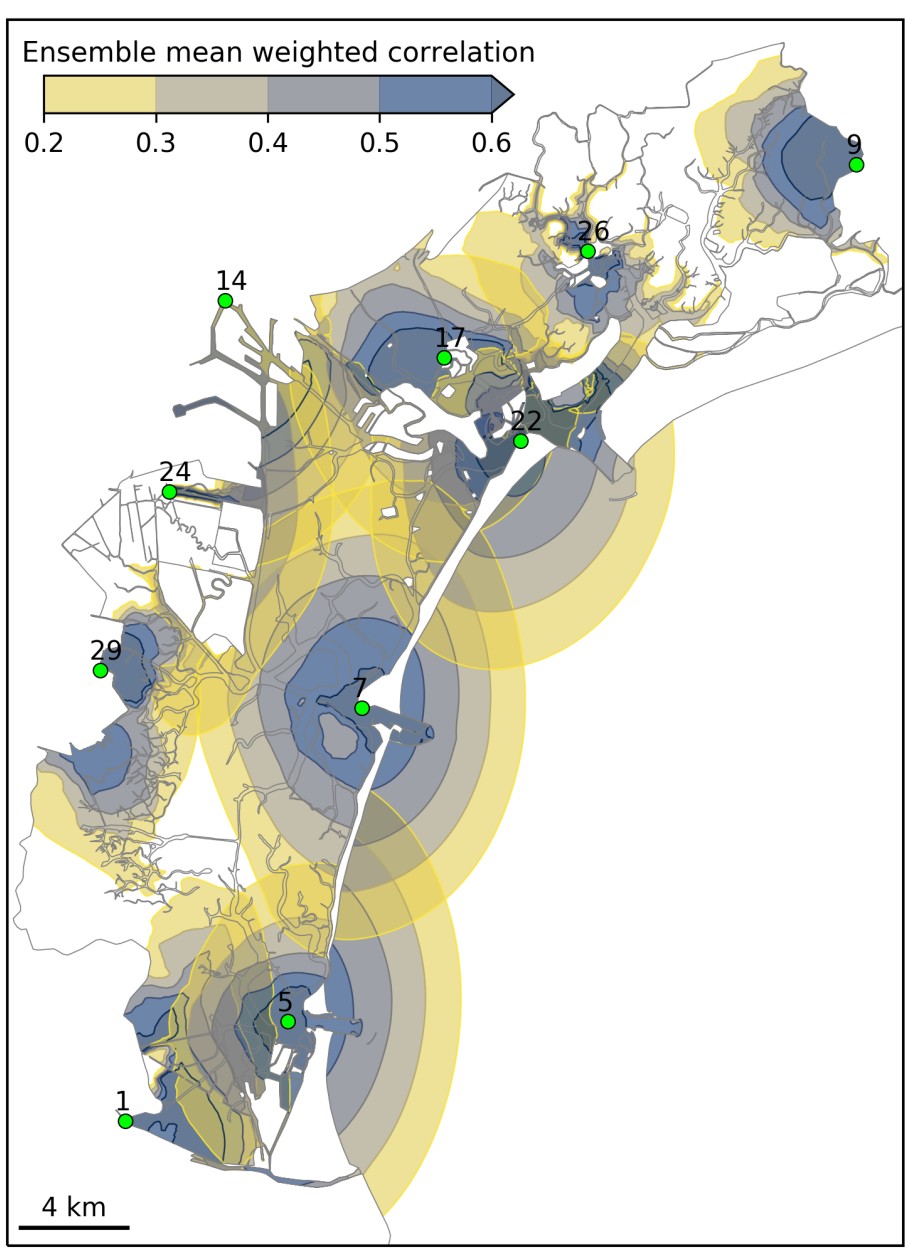

**Figure 9.** Ensemble weighted correlation (averaged over the simulation period) of the 10 monitoring stations selected using the EnSRF method.





**Table 1.** Root mean square errors (in cm) of DI and DA considering all other stations except the one for which the index is computed. The RMSEs of the control simulation are also reported.

| Station ID | Data interp. | Control sim. | DA-Nudging | DA-EnSRF |
|---|---|---|---|---|
| 1 | 4.5 | 8.3 | 5.8 | 7.5 |
| 2 | 0.8 | 4.1 | 0.9 | 2.4 |
| 3 | 3.0 | 5.2 | 2.8 | 3.4 |
| 4 | 6.5 | 4.1 | 2.5 | 3.5 |
| 5 | 1.4 | 4.1 | 1.8 | 2.1 |
| 6 | 1.5 | 4.2 | 1.4 | 2.3 |
| 7 | 3.1 | 5.2 | 1.9 | 3.1 |
| 8 | 2.4 | 5.3 | 1.4 | 2.3 |
| 9 | 7.3 | 9.5 | 3.3 | 9.4 |
| 10 | 2.5 | 4.0 | 0.9 | 2.4 |
| 11 | 3.8 | 4.3 | 2.2 | 3.0 |
| 12 | 8.5 | 4.5 | 2.5 | 4.7 |
| 13 | 3.6 | 4.6 | 0.9 | 2.3 |
| 14 | 6.0 | 6.6 | 3.9 | 2.5 |
| 15 | 2.8 | 3.7 | 2.7 | 2.6 |
| 16 | 1.9 | 4.1 | 1.0 | 1.8 |
| 17 | 2.6 | 4.5 | 1.4 | 2.1 |
| 18 | 7.3 | 4.3 | 1.1 | 2.3 |
| 19 | 2.6 | 3.6 | 0.9 | 1.5 |
| 20 | 4.2 | 3.8 | 0.7 | 1.6 |
| 21 | 2.5 | 4.1 | 0.8 | 1.4 |
| 22 | 4.3 | 3.7 | 0.9 | 2.0 |
| 23 | 2.3 | 3.4 | 2.5 | 1.9 |
| 24 | 5.2 | 28.3 | 4.4 | 7.0 |
| 25 | 8.2 | 4.9 | 2.1 | 2.8 |
| 26 | 1.1 | 4.2 | 1.3 | 2.5 |
| 27 | 6.4 | 6.6 | 1.9 | 4.0 |
| 28 | 2.8 | 5.2 | 1.7 | 3.6 |
| 29 | 3.5 | 8.5 | 5.4 | 6.2 |
| MEAN | 3.9 | 5.8 | 2.1 | 3.2 |





**Table 2.** Statistical analysis of simulated current velocity at the Lido inlet. Results are given as RMSE (root mean square error, cm s$^{-1}$), BIAS (difference between the mean of simulation results and observations, cm s$^{-1}$) and R (correlation coefficient between model results and observations).

| Simulation | RMSE | BIAS | R |
|---|---|---|---|
| Control | 14.1 | 0.6 | 0.84 |
| DA-Nudging | 15.7 | 5.3 | 0.83 |
| DA-EnSRF | 13.6 | 0.4 | 0.85 |