# Peer review of "Model-driven optimization of coastal sea observatories through data assimilation in a finite element hydrodynamic model (SHYFEM v. 7\_5\_65)"

_Geoscientific Model Development, 2020_

## Referee Comment (RC1) · Joseph Wallwork (Referee) · 1 Jun 2020

General comments

"Model-driven optimization of coastal sea observatories through data assimilation in a finite element hydrodynamic model (SHYFEM v.7_5_65)" is a well written manuscript concerned with using data assimilation to improve coastal modelling capabilities and optimise monitoring networks. The article contains a fair comparison of data interpolation (DI) and data assimilation (DA) methods, along with a further comparison of two

different DA approaches.

The Lagoon of Venice application involves a complex spatial domain with sensitive coastal dynamics. The objective of the numerical experimentation is clearly stated. Results are clearly presented in a number of attractive figures. In the conclusion, the objective is fulfilled and recommendations are made for modifying the monitoring network.

Specific comments

* I particularly liked the introduction to DI and DA philosophy given in the paragraph starting on line 37.

* Equations (6)–(7) could perhaps be introduced in a better way. There is a lot of notation all at once, some of which is not referred to in the text. For example, it would be good to elaborate on what is meant by the superscript "a" for "analysis" (although this does become clear at the end of Section 2).

* In the paragraph beginning on line 201, it would be beneficial to clarify how unforced boundary conditions are represented within the shallow water model. Forced boundaries are mentioned, but the implementation of unforced boundary conditions (for example in urban areas) is unclear. Are free-slip conditions used?

* The statement on lines 274–275 claims that "results improved at all stations". However, there appears to be one exception at station 12, where the control simulation RMSE is 4.5 but the DA-EnSRF RMSE is 4.7. This should at least be mentioned and a sentence suggesting a reason for the anomaly would be beneficial.

Technical corrections

There are a small number of typos and grammatical errors in the current manuscript:

* Line 65: "been already" -> "already been"

* Lines 70–76: Inconsistent notation "pa" vs "p_a"

* Line 222: "We run also several preliminary tests to empirically found the best cut-off distance for the local analysis" needs rewording.

* Lines 287–288: "using DA-Nudging resulted to be higher and slightly out of phase than the observations" needs rewording.

* Line 288: "interesting" -> "interestingly"

* Line 298: "composed by" -> "composed of"

* Line 312: "give" -> "gives"

---

## Short Comment (SC1) · 16 Jun 2020

Question: In the paragraph beginning on line 201, it would be beneficial to clarify how unforced boundary conditions are represented within the shallow water model. Forced boundaries are mentioned, but the implementation of unforced boundary conditions (for example in urban areas) is unclear. Are free-slip conditions used?

Response: Unforced boundaries are solid boundaries that are implemented in the model with a free slip condition. The only condition that is enforced on these boundaries are the no-flux condition through these boundaries. No-slip conditions can also be implemented by the model, however, the resolution of the numerical grid is much too coarse for these kind of condition.

We have inserted the following sentence at line 84:

At the boundaries, either water levels are prescribed at the open boundaries or the free-slip condition is implemented at solid (closed) boundaries.

---

## Referee Comment (RC2) · Joseph Wallwork (Referee) · 13 Jul 2020

Dear authors,

Thank you very much for your responses to my recommendations. I look forward to reading the updated manuscript.

---

## Referee Comment (RC3) · Anonymous Referee #2 · 10 Nov 2020

Review of

Title: Model-driven optimization of coastal sea observatories through data assimilation in a finite element hydrodynamic model (SHYFEM v.7_5_65)
Author(s): Christian Ferrarin et al.
MS No.: gmd-2020-61
MS type: Development and technical paper

The paper analyzes optimization of observational grid via analyzing the impact that assimilation of station data has on the high resolution numerical model of the Venice lagoon. Several modes of assimilation are employed to introduce data into the model. I must say I really like the idea of how DA was used in the paper. The paper is interesting, contains new insight and is well written. The figures are clear. The abstract reflects the contents well.

I recommend publication after minor revision.

Specific comments:

p3, L76: pa should be $p_a$ (_a denoting subscript)
p3, L77: \rho_q should be \rho_w
p4, L107: ""the mean is:" should probably be "the ensemble mean is:"

p7 L185: I am not sure I understand this phrase "...at which degree the obervations represent the state variable over the whole system." Can the authors please include a specific description and/or metrics by which this degree was measured?

P8 L207: should sigma be a greek letter? Why did you set it to 2 km rather than something else?

Perhaps I missed something but I still do not clearly understand how the boundary condition perturbations were generated. The paper states that 60 perturbations (gaussian, it seems?) were used as OBCs. Do I understand correctly that you used mean(A) as the open boundary conditions and then further added a constant (in space and time) perturbation to each ensemble member, where the amount of each member sea level perturbation was sampled from a gaussian $N$(mu, sigma)?

P9, L255: perhaps: "...towards the observations WHILE keeping the physical dynamics..."

p10, L298: I don't entirely see what is meant by "scalability". Can you please rephrase or clarify?

P12, L351: These correlationS…

---

## Author Comment (AC2) · 11 Nov 2020

Dear Reviewer,

We would like to thank you very much for your tireless efforts in reviewing the manuscript and for your valuables comments, which will certainly improve our work.

The original Reviewer's comments and suggestions are shown in regular typeface, while our responses are shown in italics.

[Figure]

Sincerely yours, Christian Ferrarin (On behalf of the authors)

**R2.1** General comments: The paper analyzes optimization of observational grid via analyzing the impact that assimilation of station data has on the high resolution numerical model of the Venice lagoon. Several modes of assimilation are employed to introduce data into the model. I must say I really like the idea of how DA was used in the paper. The paper is interesting, contains new insight and is well written. The figures are clear. The abstract reflects the contents well.

I recommend publication after minor revision.

*Response: We thank the reviewer for the positive comment and we improved the manuscript following all reviewer's suggestions.*

**R2.2** p3, L76: $pa$ should be $p_a$ ($_a$ denoting subscript)

*Response: Corrected.*

**R2.3** p3, L77: $\rho_q$ should be $\rho_w$

*Response: Corrected.*

**R2.4** p4, L107: "the mean is:" should probably be "the ensemble mean is:"

*Response: Corrected.*

**R2.5** p7 L185: I am not sure I understand this phrase "... at which degree the observations represent the state variable over the whole system." Can the authors please include a specific description and/or metrics by which this degree was measured?

*Response: As explained in section 3.1, for both Data Interpolation and Data Assimilation experiments the metric used to evaluate the representativeness of the method in describing the state variable over the system is the root mean square*

*error. RMSE is evaluated in the station not considered in the DI or DA compu-*
*tations. The evaluation procedure was repeated for each monitoring station and*
*the results are reported in Table 1.*

**R2.6** P8 L207: should sigma be a greek letter? Why did you set it to 2 km rather than
something else?

*Response: We corrected the sigma Greek letter. As specified at the beginning of*
*the Results section, all parameters ($\sigma$, $\tau$, cut-off distance for the local analysis)*
*were manually defined through trial and error calibration process and evaluating*
*the goodness-of-fit of the water level RMSE in the DA-Nudging and DA-EnSRF*
*base simulations.*

**R2.7** Perhaps I missed something but I still do not clearly understand how the bound-
ary condition perturbations were generated. The paper states that 60 perturba-
tions (gaussian, it seems?) were used as OBCs. Do I understand correctly that
you used mean(A) as the open boundary conditions and then further added a
constant (in space and time) perturbation to each ensemble member, where the
amount of each member sea level perturbation was sampled from a gaussian
N(mu, sigma)?

*Response: We concur with the reviewer that the perturbation terms were not*
*properly described. At each timestep ($t$), a random vector ($r$) of $N$ perturbations*
*is computed from a Gaussian distribution (with mean 0 and standard deviation of*
*30 cm) as:*

$$r(t, n) = cos(2\pi r_2(t, n))\sqrt{-2log(r_1(t, n) + \epsilon)} \tag{1}$$

*with $n$ the number of the ensemble member ($1, N$), $r_1$ and $r_2$ random vectors and*
*$\epsilon$ a very small number.*

*The perturbation vector $p$ at time $t$ is computed using the random vector and the*

[Figure]

*perturbation vector at the previous time ($t_{-1}$):*

$$p(t, n) = \alpha p(t_{-1}, n) + \sqrt{1 - \alpha^2} r(t, n) \tag{2}$$

*with $\alpha = 1 - (t - t_{-1})/\tau$ and $\tau$ the decay time (2 days in our case). Then the new perturbation is stored for the next time step. This type of perturbations are classified as red noise.*

*We modified the manuscript to clarify the methodology adopted in this study. The text now reads: "We used 60 perturbations for the sea-level boundary condition (member 0 is unperturbed) taken from a Gaussian distribution with a zero mean and a standard deviation set to 30 cm. This value was found empirically, in order to have a good spread at the boundary, which is then propagated to the variables computed by the model. As asserted, the perturbations are centred, having a null mean, and correlated in time. To do this, each perturbation at time $t$ is obtained from a weighted average of a new perturbation and of the one at time $t - 1$. This type of perturbations are classified as red noise and in the present case we used a decay time of two days."*

**R2.8** P9, L255: perhaps: "...towards the observations WHILE keeping the physical dynamics..."

*Response: Corrected.*

**R2.9** p10, L298: I don't entirely see what is meant by "scalability". Can you please rephrase or clarify?

*Response: We are referring to the computing scalability of the DA-EnSRF procedure on multiprocessor computers. The sentence has been rephrased as follow: "So in this case the simulations are 26535, but the computing scalability is high since the 61 simulations of the ensemble are independent and can be parallelized on multiple CPUs computers.*

**R2.10** P12, L351: These correlationS...

*Response: Corrected.*

---

## Author Response (AR1)

**Responses to the Reviewers' Comments and Suggestions**

Journal: Geoscientific Model Development (GMD)

Manuscript number: gmd-2020-61

Manuscript title: Model-driven optimization of coastal sea observatories through data assimilation in a finite element hydrodynamic model (SHYFEM v.7_5_65)

We would like to thank the Reviewers for their valuable comments and effort to improve the manuscript. We have responded to all comments as can be seen in the following list. We believe that with these revisions the manuscript has been improved and we hope that it is now ready for publication.

The original Reviewers' comments and suggestions are shown in regular typeface, while our responses are shown in italics. The line and figures numbers we use refer to the revised document.

**Response to Reviewer #1**

**R1.1** General comments: "Model-driven optimization of coastal sea observatories through data assimilation in a finite element hydrodynamic model (SHYFEM v.7_5_65)" is a well written manuscript concerned with using data assimilation to improve coastal modelling capabilities and optimise monitoring networks. The article contains a fair comparison of data interpolation (DI) and data assimilation (DA) methods, along with a further comparison of two different DA approaches. The Lagoon of Venice application involves a complex spatial domain with sensitive coastal dynamics. The objective of the numerical experimentation is clearly stated. Results are clearly presented in a number of attractive figures. In the conclusion, the objective is fulfilled and recommendations are made for modifying the monitoring network.

*Response: We appreciate the comments and we improved the manuscript following all reviewer's suggestions.*

**R1.2** I particularly liked the introduction to DI and DA philosophy given in the paragraph starting on line 37.

*Response: We thank the reviewer for this positive comment.*

**R1.3** Equations (6)-(7) could perhaps be introduced in a better way. There is a lot of notation all at once, some of which is not referred to in the text. For example, it would be good to elaborate on what is meant by the superscript "a" for "analysis" (although this does become clear at the end of Section 2).

*Response: We concur with the reviewer that some of the mathematical passages were poorly explained. In the revised manuscript, we improved the explanation of the mentioned equations by adding more details on the different terms (lines 110-132).*

**R1.4** In the paragraph beginning on line 201, it would be beneficial to clarify how unforced boundary conditions are represented within the shallow water model. Forced boundaries

are mentioned, but the implementation of unforced boundary conditions (for example in urban areas) is unclear. Are free-slip conditions used?

*Response: Unforced boundaries are solid boundaries that are implemented in the model with a free slip condition. The only condition that is enforced on these boundaries are the no-flux condition through these boundaries. No-slip conditions can also be implemented by the model, however, the resolution of the numerical grid is much too coarse for these kind of condition.*

*We have inserted the following sentence at lines 84-85: "At the boundaries, either water levels are prescribed at the open boundaries or the free-slip condition is implemented at solid (closed) boundaries".*

**R1.5** The statement on lines 274-275 claims that "results improved at all stations". However, there appears to be one exception at station 12, where the control simulation RMSE is 4.5 but the DA-EnSRF RMSE is 4.7. This should at least be mentioned and a sentence suggesting a reason for the anomaly would be beneficial.

*Response: We thank the reviewer for highlighting this anomaly which was due to a type-setting error in the LaTeX file. The correct value of 3.7 is now reported in Table 1.*

**R1.6** There are a small number of typos and grammatical errors in the current manuscript.

*Response: We corrected all reported typos and grammatical errors.*

**Response to Reviewer#2**

**R2.1** General comments: The paper analyzes optimization of observational grid via analyzing the impact that assimilation of station data has on the high resolution numerical model of the Venice lagoon. Several modes of assimilation are employed to introduce data into the model. I must say I really like the idea of how DA was used in the paper. The paper is interesting, contains new insight and is well written. The figures are clear. The abstract reflects the contents well.

I recommend publication after minor revision.

*Response: We thank the reviewer for the positive comment and we improved the manuscript following all reviewer's suggestions.*

**R2.2** p3, L76: $pa$ should be $p_a$ ($_a$ denoting subscript)

*Response: Corrected.*

**R2.3** p3, L77: $\rho_q$ should be $\rho_w$

*Response: Corrected.*

**R2.4** p4, L107: "the mean is:" should probably be "the ensemble mean is:"

*Response: Corrected.*

**R2.5** p7 L185: I am not sure I understand this phrase "... at which degree the observations represent the state variable over the whole system." Can the authors please include a specific description and/or metrics by which this degree was measured?

*Response: As explained in section 3.1, for both Data Interpolation and Data Assimilation experiments, the metric used to evaluate the representativeness of the method in describing the state variable over the system is the root mean square error. RMSE is evaluated in the station not considered in the DI or DA computations. The evaluation procedure was repeated for each monitoring station and the results are reported in Table 1.*

**R2.6** P8 L207: should sigma be a greek letter? Why did you set it to 2 km rather than something else?

*Response: We corrected the sigma Greek letter. As specified at the beginning of the Results section (lines 247-249), all parameters ($\sigma$, $\tau$, cut-off distance for the local analysis) were manually defined through trial and error calibration process and evaluating the goodness-of-fit of the water level RMSE in the DA-Nudging and DA-EnSRF base simulations.*

**R2.7** Perhaps I missed something but I still do not clearly understand how the boundary condition perturbations were generated. The paper states that 60 perturbations (gaussian, it seems?) were used as OBCs. Do I understand correctly that you used mean(A) as the open boundary conditions and then further added a constant (in space and time) perturbation to each ensemble member, where the amount of each member sea level perturbation was sampled from a gaussian N(mu, sigma)?

*Response: We concur with the reviewer that the perturbation terms were not properly described. At each timestep (t), a random vector (r) of N perturbations is computed from a Gaussian distribution (with mean 0 and standard deviation of 30 cm) as:*

$$r(t,n) = cos(2\pi r_2(t,n))\sqrt{-2log(r_1(t,n) + \epsilon)} \qquad (1)$$

*with n the number of the ensemble member $(1, N)$, $r_1$ and $r_2$ random vectors and $\epsilon$ a very small number.*

*The perturbation vector p at time t is computed using the random vector and the perturbation vector at the previous time $(t_{-1})$:*

$$p(t,n) = \alpha p(t_{-1}, n) + \sqrt{1 - \alpha^2}r(t,n) \qquad (2)$$

*with $\alpha = 1 - (t - t_{-1})/\tau$ and $\tau$ the decay time (2 days in our case). Then the new perturbation is stored for the next time step. This type of perturbations are classified as red noise.*

*We modified the manuscript to clarify the methodology adopted in this study. The text now reads (lines 220-225): "We used 60 perturbations for the sea-level boundary condition (member 0 is unperturbed) taken from a Gaussian distribution with a zero mean and a standard deviation set to 30 cm. This value was found empirically, in order to have a good spread at the boundary, which is then propagated to the variables computed by the model. As asserted, the perturbations are centred, having a null mean, and correlated in time. To do this, each perturbation at time t is obtained from a weighted average of a new perturbation and of the one at time $t - 1$. This type of perturbations are classified as red noise and in the present case we used a decay time of two days."*

**R2.8** P9, L255: perhaps: "...towards the observations WHILE keeping the physical dynamics..."

*Response: Corrected.*

**R2.9** p10, L298: I don't entirely see what is meant by "scalability". Can you please rephrase or clarify?

*Response: We are referring to the computing scalability of the DA-EnSRF procedure on multiprocessor computers. The sentence has been rephrased as follow (lines 308-309): "So in this case the simulations are 26,535, but the computing scalability is high since the 61 simulations of the ensemble are independent and can be parallelized on multiple CPUs computers.*

**R2.10** P12, L351: These correlationS...

*Response: Corrected.*

[revised manuscript text omitted]